# Peptidomic Characterization and Amino Acid Availability after Intake of Casein vs. a Casein Hydrolysate in a Pig Model

**DOI:** 10.3390/nu15051065

**Published:** 2023-02-21

**Authors:** Pablo Jiménez-Barrios, Laura Sánchez-Rivera, Daniel Martínez-Maqueda, Yann Le Gouar, Didier Dupont, Beatriz Miralles, Isidra Recio

**Affiliations:** 1Instituto de Investigación en Ciencias de la Alimentación, CIAL (CSIC-UAM, CEI UAM+CSIC), Nicolás Cabrera, 9, 28049 Madrid, Spain; 2Institut National de Recherche Pour l’Agriculture, l’Alimentation et l’Environnement (INRAE) STLO Agrocampus Ouest, 35042 Rennes, France

**Keywords:** casein, hydrolysate, duodenal digest, peptidomics, digestion kinetics

## Abstract

It is known that casein hydrolysis accelerates gastrointestinal transit in comparison to intact casein, although the effect of the protein hydrolysis on the composition of the digests is not fully understood. The aim of this work is to characterize, at the peptidome level, duodenal digests from pigs, as a model of human digestion, fed with micellar casein and a previously described casein hydrolysate. In addition, in parallel experiments, plasma amino acid levels were quantified. A slower transit of nitrogen to the duodenum was found when the animals received micellar casein. Duodenal digests from casein contained a wider range of peptide sizes and a higher number of peptides above five amino acids long in comparison with the digests from the hydrolysate. The peptide profile was markedly different, and although β-casomorphin-7 precursors were also found in hydrolysate samples, other opioid sequences were more abundant in the casein digests. Within the same substrate, the evolution of the peptide pattern at different time points showed minimal changes, suggesting that the protein degradation rate relies more on the gastrointestinal location than on digestion time. Higher plasma concentrations of methionine, valine, lysine and amino acid metabolites were found in animals fed with the hydrolysate at short times (<200 min). The duodenal peptide profiles were evaluated with discriminant analysis tools specific for peptidomics to identify sequence differences between both substrates that can be used for future human physiological and metabolic studies.

## 1. Introduction

Milk is an excellent source of highly digestible proteins in human nutrition. It is well known that the speed of protein digestion and amino acid absorption is affected by the protein nature and the technological treatment applied, and protein digestion kinetics modulate the metabolic response. While the casein fraction of milk, so-called slow protein, coagulates in the stomach and promotes postprandial protein deposition, whey proteins induce a fast increase in plasma amino acids, protein synthesis, and oxidation [1]. Small changes in processing have also been found to affect the kinetics of protein digestion. For instance, dairy gels differing in the mode of coagulation (acidification vs. renneting) cause different compositions of porcine duodenal effluents and plasma amino acids [2].

Enzymatic hydrolysis is increasingly used in the dairy sector for the design of ingredients for infant formulas [3]. In addition, casein hydrolysates have demonstrated to stimulate insulin secretion and to show glycemic control properties in both animals and humans [4,5]. A diet containing hydrolyzed casein has been reported to enhance upper gastrointestinal transit and to increase ileal microbial carbohydrate metabolism in comparison with an intact casein-based diet [6]. However, the influence of the enzymatic hydrolysis of the casein fraction of milk on gastric emptying and composition of the gastrointestinal contents is not fully understood. A previous study with aged rats found that casein or whey hydrolysates do not alter gastrointestinal transit compared with their respective intact counterparts. The slowed gastrointestinal transit of casein was reversed with opioid inhibitors, suggesting the involvement of opioid receptors [7]. Similarly, protein hydrolysis has been described to reduce the gastrointestinal transit time in preterm infants [8]. In rats, this effect was also attributed to the reduction of casein-derived opioid receptor agonists [9].

The advances in peptidomic techniques based on mass spectrometry (MS) have allowed for study of the complex protein digestome at various regions of the gastrointestinal tract [10,11]. These techniques have widely been used to characterize the large variety of peptides generated by the gastrointestinal proteases and peptidases, mainly in vitro [12,13,14,15]. However, in recent years, the characterization of gastrointestinal effluents from human or pig, as an accepted model of human digestion, is building a valuable peptide repertoire, especially after oral administration of milk proteins, to be used in physiological and metabolic studies or to validate in vitro digestion protocols [16,17,18].

The aim of this work is to characterize the peptidome of duodenal digests obtained in cannulated pigs after the oral administration of micellar casein and a casein hydrolysate. Different time points from 5 to 150 min were considered to evaluate kinetics of protein digestion in the duodenum. In addition, in parallel experiments, quantitative analysis of plasma amino acids was performed to assess the impact of protein hydrolysis on amino acid availability. Because pig is recognized as an excellent model of the human gastrointestinal tract, the results offer new information about the faster digestion and quicker availability of amino acids of protein hydrolysates used in human nutrition.

## 2. Materials and Methods

### 2.1. Animal Experiment

All procedures were in accordance with the European Community guidelines for the use of laboratory animals (L358-86/609/EEC). The local committee for ethics in animal experimentation approved the study. The animals were 6 Large White × Landrace × Piétrain pigs of about 40 kg live weight. Two weeks before starting the trials, they were surgically fitted with a T-shaped cannula in the duodenum (10 cm downstream from the pylorus). Catheterization of the external jugular vein was also performed. Following the operation, the animals were housed in individual slatted pens within a ventilated room with controlled temperature (21 °C). During convalescence following the surgical procedure and also between the sampling days, pigs were fed with 800 g/d of a pig feed concentrate (Cooperl Arc Atlantique^®^, Pelstan, France) containing 16% proteins, 1% fat, 4% cellulose and 5% mineral matter and had free access to water.

The test meals were commercial micellar casein (Prodiet 85B, batch no. 141179) purchased from Ingredia S. A., Arras, France, and a pepsin hydrolysate from this casein. The hydrolysate was prepared with food grade pepsin (P389P/3 Biocatalyst 3000 U/mg) at 2% enzyme/substrate ratio (*w*/*w*), added at 0 and 3 h during a hydrolysis time of 6 h at 37 °C, as previously reported [19]. The nitrogen content of casein and the hydrolysate was 77.8 and 77.5%, respectively, as determined by Kjeldahl.

The collection of duodenal effluents and blood samples was performed in separate assays. In each one, both dairy products were randomly tested on each animal. Within a period (2 weeks), the days of sampling were separated by at least 2 day. Test meals, casein and the casein hydrolysate (129 g of powder, on the basis of protein content) were reconstituted in water (250 mL) and offered to overnight fasted pigs for 10 min. The pigs had no access to water until 4 h after the meal delivery. Duodenum effluents were collected in plastic bottles 15 min before and 5, 10, 15, 20, 30, 45, 60, 90, 120, and 150 min after ingestion of test meal. The sampling was stopped when 40 mL was collected or after a maximum of 3 min of sampling time. A protease inhibitor (Pefabloc^®^ at 2.5 mM final concentration) was previously added to the sample tubes. Collected effluents were weighed, freeze-dried and stored at −20 °C.

Blood sampling was performed at 15 min before and 5, 20, 45 min, 1, 2, 3, 4 and 6 h after ingestion of the casein or its hydrolysate. Blood samples (5 mL) were collected in heparinized-lithium tubes where a protease inhibitor cocktail for plasma samples was previously added (1 tablet/10 mL blood) (Complete Mini, EDTA-free, Roche Farma, Madrid, Spain). Then, centrifugation at 3000× *g* 4 °C for 10 min was performed, and the supernatant was frozen with liquid nitrogen and stored at −80 °C. Amino acids and related metabolites in plasma were determined by ion-exchange chromatography with ninhydrine at the Diagnosis of Molecular Diseases Centre from the Autonomous University of Madrid. Briefly, 2 mL of plasma was deproteinized with 200 μL 10% trichloroacetic acid. After incubation for 15 min on ice, the samples were centrifuged for 10 min at 1500× *g* and 4 °C, and 150 μL of supernatant was collected. The samples were placed in an autosampler protected from light, and 20 μL was injected.

### 2.2. Nitrogen Analysis and SDS-PAGE

Nitrogen content of duodenal effluents was determined by elemental analysis in a LECO chns-932 analyzer at the Elemental analysis unit from the Interdepartmental Investigation Service from the Autonomous University of Madrid. Freeze-dried digests were adjusted to 1 mg of protein/mL of sample buffer (tris-HCL 0.05 M pH 6.8, SDS 1.6% *w:v*, glycerol 8% *v*:*v*, bromophenol blue 0.002% *w:v*, and β-mercaptoethanol 2% *v*:*v*) based on the nitrogen content, and the SDS-PAGE was conducted in Precast Criterion XT 12% Bis-Tris gels (Bio-Rad, Richmond, CA, USA) using the running buffer (19:1, *v*/*v* water/XT MES 20×; Bio-Rad Laboratories, Richmond, CA, USA). A molecular weight marker (Precision Plus ProteinTM Unstained standard, Bio-Rad Laboratories, Richmond, CA, USA) was used. A prior running at 100 V for 5 min was performed, and then, samples were run at 150 V for 45 min. Gel was rinsed with water, then stained with Coomasie Blue (Instant blue; Expedeon, Swavesey, UK) for 30 min, and rinsed again with water for 10 min.

### 2.3. Peptidomic Analysis

Tandem mass spectrometry (HPLC-MS/MS) was conducted on an Agilent 1100 HPLC system (Agilent Technologies, Waldbron, Germany) coupled to an Esquire 3000 ion trap mass spectrometer (Bruker Daltonics, Bremen, Germany) as previously described [20]. A linear gradient from 0 to 45% solvent B (100:0.1, *v*/*v* acetonitrile:formic acid) for 120 min on a Mediterranean Sea C18 column, 150 × 2.1 mm (Teknokroma, Barcelona, Spain) was used at 0.2 mL/min. The freeze-dried digests were reconstituted in solvent A (100:0.1, *v*/*v* water:formic acid) and the supernatants after centrifugation at 13,000× *g*, for 10 min were analyzed. Spectra were recorded over the mass/charge (*m/z*) range 100–1200, and the target mass was set at 900 *m*/*z*. Data processing was conducted using the software Data Analysis, version 4.0 and Biotools, version 3.2 from Bruker Daltonics. MASCOT V2.4 (MatrixScience) was used for peptide identification with the use of homemade databases of major genetic variants of bovine caseins and porcine endogenous proteins. The parameters selected were: no enzyme, no post-translational modifications, and peptide tolerance 0.1% and 0.5 Da for MS and MS/MS, respectively. A manual revision for each identification was performed, regardless of the *p* value.

To carry out most peptidomic analysis steps, R scripts were used. R scripts are some lines of code using R programing language. Peptide size distribution of all samples was calculated. Peptide sequences were divided by substrate and then subdivided by size. Raw numbers and their percentages were shown using the R package “ggplot2” version 3. 4.0 [21]. Peptides previously grouped by substrate were analyzed using the R package “ggseqlogo” version 0.1 to obtain the SeqLogo figure [22]. The sequence logo shows the probability to find a particular amino acid at the *N*-terminal (1, 2, 3) or *C*-terminal (−3, −2, −1) end of the sequence. In the case of five-length peptides, the amino acid in the 3rd position at both terminal ends was the same.

### 2.4. Statistical Analysis

Principal components analysis (PCA) was performed using the amino acids within the identified peptides as variables. For each amino acid and each time point, the value was calculated as the sum of intensities of all overlapping peptides. A hierarchical clustering tree of samples using the intensity of the peptides was built by PermutMatrix software version 1.9.3.0 [23]. Peptides present in at least two time points were used. Blood samples were analyzed using GraphPad Prism software version 8.0.2. Differences between substrates over time were found through a two-way ANOVA, followed by the Bonferroni test. The results were graphed with the R package “ggplot2”version 3.4.0 [21].

## 3. Results and Discussion

Duodenal contents after oral administration of casein and its hydrolysate were collected and freeze-dried until analysis, and the nitrogen content of the dried effluents is shown in Figure 1. Duodenal digests from casein had higher nitrogen content than those from the hydrolysate at 90 and 150 min, showing its emptying over a more extended period, compared to the hydrolysate. Although at short time points (<30 min), hydrolysate effluents showed higher protein content than casein ones, due to the high variability between animals, differences did not reach significance. Because the effluents were immediately frozen in liquid nitrogen and freeze-dried, flow rates of liquid effluents could not be measured, and differences in the nitrogen content between substrates could be diminished. Previous reports in aged rats have shown unaltered gastrointestinal transit with hydrolyzed casein compared with its intact counterpart [7]. Casein was emptied from the stomach mainly in the form of peptides, although in some time points (15, 45, 60 min), slight bands of intact casein fractions were visible by SDS-PAGE (Appendix A). In addition, because the casein substrate contained a small amount of whey proteins (β-lactoglobulin and α-lactalbumin), bands corresponding to these proteins were also detected up to 30 min after oral administration. Due to their high solubility, whey proteins are rapidly emptied from the stomach [1], and the net disappearance of β-lactoglobulin from human upper jejunum was reported to be faster than for casein [24]. Similarly, Barbé and co-workers found β-lactoglobulin in porcine duodenum after the administration of raw liquid milk [25]. In contrast to casein, the electrophoretic protein profile of the porcine duodenal effluents after the hydrolysate administration slightly varied with time, and all protein material detected was below 10 kDa.

The peptide fraction of the duodenal effluents was characterized by HPLC-MS/MS. A total of 3863 milk-derived peptides, around 400 peptides per time point, were identified in porcine duodenum after the administration of casein, and a lower number, ca. 300 per time point, was found after the administration of the hydrolysate (Table 1). This lower number in the case of the hydrolysate can be due to the limitation of our mass spectrometry conditions to identify peptides shorter than five amino acids. In addition, Table 1 shows the number of common peptides found in all animals (*n* = 6) to illustrate the inter-individual variability found. When using casein as feeding, the peptide profile showed a large variety of peptide sizes where ca. 25% were above 10 amino acids long. However, as expected, the peptides longer than 10 amino acids were below 10% when the protein was administered in the form of hydrolysate (Figure 2). This peptide size distribution matches with the information provided by SDS-PAGE (Appendix A). The most abundant peptides identified in both substrates corresponded to peptides between six and nine amino acids, but this can be caused by the selectivity of the MS method employed to detect peptides within this size range. Despite this, the peptide profile found in the duodenum after the administration of both substrates was notably different (Figure 3A,B). After casein administration, the most abundant β-casein-derived peptides belonged to the *C*-terminal domain of the protein, i.e., 193–209, while peptides in regions 58–80, and 115–126 from β-casein are specifically overrepresented in the duodenal effluents from the hydrolysate (Figure 3A). Other authors have also reported the resistance of the *C*-terminal region of β-casein to gastric pepsin both in simulated gastric digestions [26,27] and in pig duodenum [18]. The slower gastrointestinal transit of casein in comparison with soy has been attributed to the involvement of opioid peptides since it was reversed with opioid inhibitors [7]. Although some studies have found an accelerated gastrointestinal transport with hydrolyzed protein infant formula due to the reduction of opioid agonists [8,9], the presence of opioid sequences depends on the hydrolysis conditions (enzyme specificity, time…). Under our hydrolysis conditions, the regions corresponding to β-casein opioid peptides, i.e., β-casomorphin-7 (^60^YPFPGPI^66^) [28] and neocasomorphin-6 (^114^YPVEPF^119^) [29] were preserved, and these peptides or precursors thereof comprised highly overrepresented regions in the duodenal effluents from the hydrolysate. For the other main casein fraction, α_s1_-casein, differences in the duodenal peptide profile between substrates were less marked (Figure 3B), but the hydrolysate effluents showed a higher abundancy of regions 24–33, 166–172, and 180–196. The domains corresponding to previously reported opioid peptides (α-casomorphins), i.e., α_s1_-casein ^90^RYLGYLE^96^ [30] and ^144^YFYPEL^149^ [31], were more abundant in casein duodenal effluents than in hydrolysate ones. Other α_s1_-casein abundant domains, such as 24–33, were common for both substrates. The compared peptide profiles for α_s2_-casein and κ-casein are shown in Appendix A, where clear differences between both substrates were also found.

In order to illustrate the evolution of the peptide profile with time, peptides were also aligned within the sequence of the main casein fractions, β-casein and α_s1_-casein, and represented in the form of heatmaps (Appendix A). These graphs are based in the appearance frequency of each amino acid without taking into account the peptide intensity. Again, the *C*-terminal region of β-casein corresponded to the domain where a higher number of peptides was identified after casein administration, but for hydrolysate effluents, regions 70–100 and 115–125 were especially rich in peptides. Within the same substrate, differences in the peptide profile with time were less relevant, concluding that protein digestion occurs along the gastrointestinal tract, and for a given part, protein degradation is similar. Similar results were previously reported in human jejunum [16,17] or in different regions of the porcine gastrointestinal tract [18]. In addition to milk-derived peptides, the HPLC-MS/MS analysis of the effluents allowed us to identify several peptides from endogenous proteins (Appendix A). The main identified proteins corresponded to endogenous enzymes: pepsin, chymotrypsin, carboxypeptidases, dipeptidyl peptidase or lipases, but also other peptides belonged to the sequence of gastrointestinal hormones, mucins or ubiquitous proteins, such as albumin.

To compare the peptide composition of the effluents for both substrates at different time points in a holistic way, different techniques of discriminant analysis for peptidomics were used. Figure 4 shows the principal component analysis built with the peptide intensity values used in Figure 3 and Appendix A (including β-, α_s1_-, α_s2_, and κ-casein). The first component explained 41.6% of the variance while PC2 reached 22.1%. The peptide intensity profiles at the different time points were clearly grouped per substrate, demonstrating the differences shown in Figure 3 and the Appendix A. In addition, effluents obtained at longer time points, i.e., 90, 120 and 150 min, were separated from those at shorter times, and this effect was observed in both substrates, although it was more notable for casein duodenal effluents.

The free software PermutMatrix was used to perform a graphical analysis of the peptide profiles based on sequence and intensity. The dendrogram in Figure 5 shows the clustering of samples per substrate, casein and hydrolysate. In addition, this analysis allows for the identification of the peptide differences in those regions overrepresented in each substrate. For instance, β-casomorphin precursors with valine or tyrosine at the *N*-terminal were highly detected in the effluents from the hydrolysate, but specific sequences, such as ^57^SLVYPFPGPI^66^ or ^58^LVYPFPGPI^66^, were more abundant in casein samples (Figure 5B). Similarly, β-casein regions containing the antihypertensive tri-peptides IPP and VPP [32] were overrepresented in the hydrolysate duodenal digests, but several related peptides found in the casein samples showed a different cleavage pattern with proline–proline at the *N*-terminal end (Figure 5C). The peptide known as neocasomorphin ^114^YPVEPF^119^ was found in duodenal digests from both casein and hydrolysate, although several peptides lacking the *N*-terminal tyrosine were especially abundant in the hydrolysate samples (Figure 5D). Because the presence of tyrosine at the *N*-terminal end is a structural requirement for opioid receptor agonists, the loss of this amino acid in the hydrolysate digests suggested a decrease in their opioid effect. Other regions where differences between substrates were notable corresponded to α_s1_-casein 154–173 (Figure 5E) and the α_s1_-casein *N*-terminal region (Figure 5F).

The analysis of the cleavage positions for both samples was carried out with ggseqlogo to visualize the most likely amino acid found at each of the three *N*- and *C*-terminal positions for both substrates (Appendix A). While for casein digests, the most probably *N*-terminal residue is a tyrosine, for hydrolysate digests, it is proline followed by tyrosine. Similarly, the most abundant *C*-terminal tripeptide is PPL and PQQ for casein and hydrolysate, respectively.

Plasma amino acids were also quantified over time of digestion. In addition to a baseline sample, blood was taken 5, 20, 45, 60, 120, 180, 240 and 360 min after test meal intake. As expected, a greater increase in total amino acids in plasma was observed with the hydrolysate from 45 to 240 min (Appendix A), but the values converged at 360 min. Previous reports in humans have described a faster rate of absorption with hydrolyzed casein than with intact casein, although similar whole-body postprandial retention of dietary nitrogen was observed [33]. Moreover, hydrolyzed casein did not enhance ileal endogenous protein losses in humans when compared with the parent protein [34]. When considering the dairy matrix of iso-protein products in the postprandial blood amino acid concentrations in human volunteers, the effect of the protein structure was evidenced as extended response curves with a micellar casein isolate as opposed to a whey protein isolate and yoghurt [35].

Figure 6 shows the determinations of selected amino where significant differences were shown at short times in the case of methionine, lysine or phenylalanine, peaking at 45 min. Other amino acids and metabolites are shown in the Appendix A. Differences in the 2–3 h post-ingestion period were observed for valine, isoleucine, proline, and two metabolites: α-aminoadipic and α-aminobutyric acids. These values are in accordance with previous results in humans [33] where indispensable amino acids were significantly higher at short times after hydrolyzed casein intake compared to intact casein.

## 4. Conclusions

Our results confirm the emptying of intact casein over a more extended period, compared to casein hydrolysate. Discriminant analysis showed a markedly different peptide profile of duodenal digests for all casein fractions between both substrates. In the opioid peptide β-casomorphin-7, precursors with differing terminal ends were observed, with a prevalence of the VYPF- and YPF-ends in the hydrolysate, whereas some longer precursors were more abundant in the casein digests. For other opioid peptides, such as neocasomorphin ^114^YPVEPF^119^, an inactive form lacking the *N*-terminal tyrosine, prevailed in the hydrolysate samples. The peptide profile at different time points within the same substrate was similar, suggesting that the degree of protein degradation of the material reaching the duodenum is similar. Finally, the amino acid concentrations found in plasma confirmed a more rapid absorption in hydrolysate-fed animals, with several branched-chain and other hydrophobic amino acids peaking at shorter times than casein. These results can be applied to dairy ingredients formulated as hydrolysates or partly hydrolyzed products, such as infant formula or products used for immune, metabolic or hormonal effects.

## Figures and Tables

**Figure 1 nutrients-15-01065-f001:**
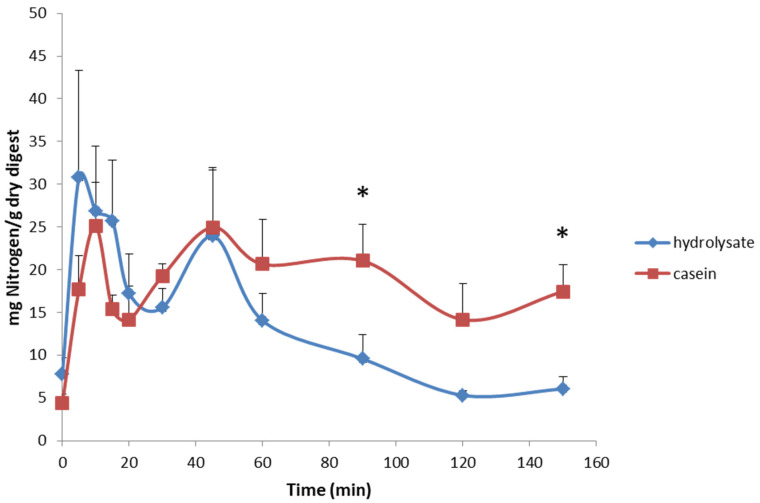
Average nitrogen content of pig duodenal digests (*n* = 6) over collection times at basal (−15 min), 5, 10, 15, 20, 30, 45, 60, 90, 120, and 150 min after intake of casein or hydrolysate. * indicates statistical difference (*p* < 0.05) by *t* test mean comparison.

**Figure 2 nutrients-15-01065-f002:**
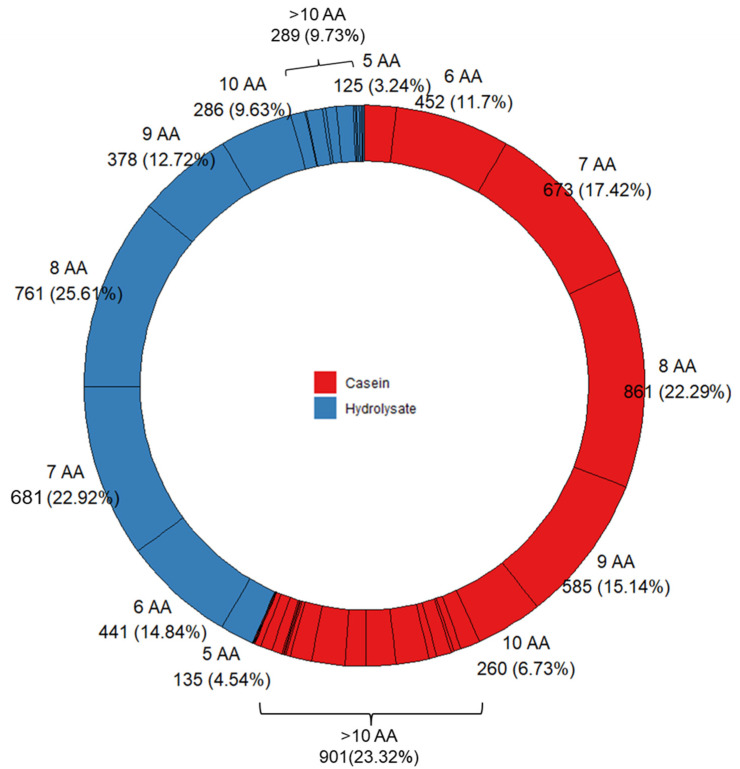
Distribution by peptide size of numbers of identified peptides derived from α_s1_-_,_ α_s2_-, β-, and κ-casein in pig duodenal digests (*n* = 6) at basal (−15 min), 5, 10, 15, 20, 30, 45, 60, 90, 120, and 150 min after intake of casein or hydrolysate.

**Figure 3 nutrients-15-01065-f003:**
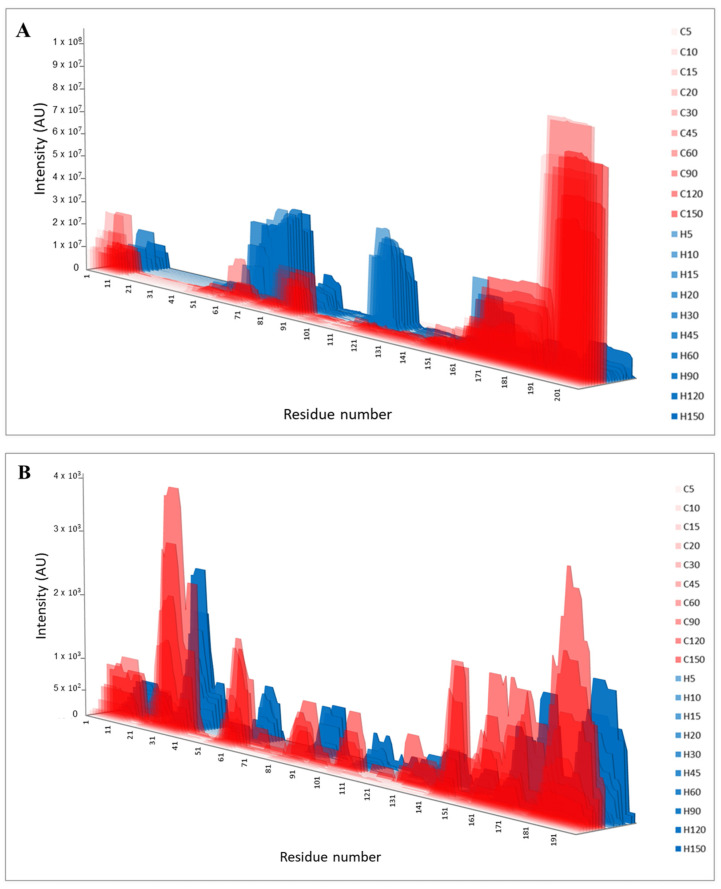
Identified peptides in pig duodenal digests (*n* = 6) after intake of casein (C) or hydrolysate (H) over collection time represented by area graphs. The height (*Y*-axis) is proportional to the sum of the identified peptide intensities. Profile peptides from β-casein (**A**) and α_s1_-casein (**B**).

**Figure 4 nutrients-15-01065-f004:**
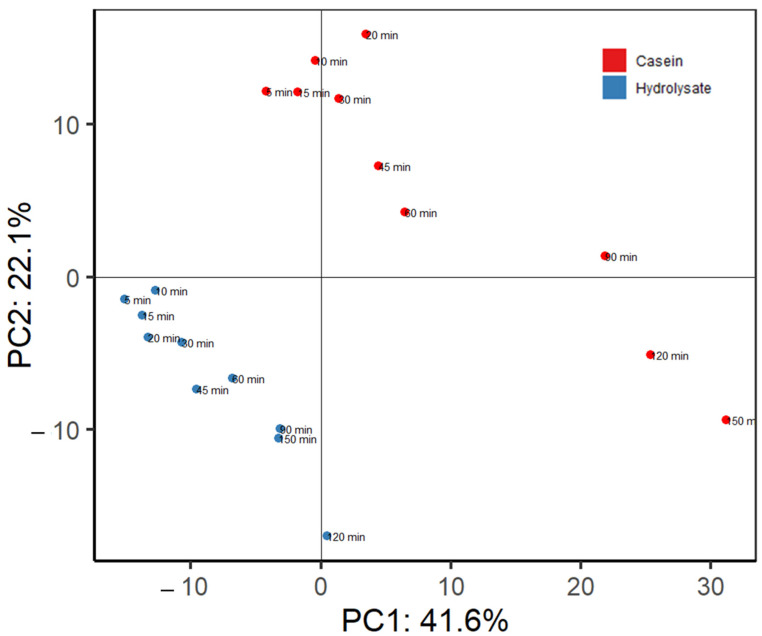
PCA analysis using, at each time point, the sum of intensities of all overlapping peptides identified in pig duodenal digests (*n* = 6) using each comprised amino acid as a variable.

**Figure 5 nutrients-15-01065-f005:**
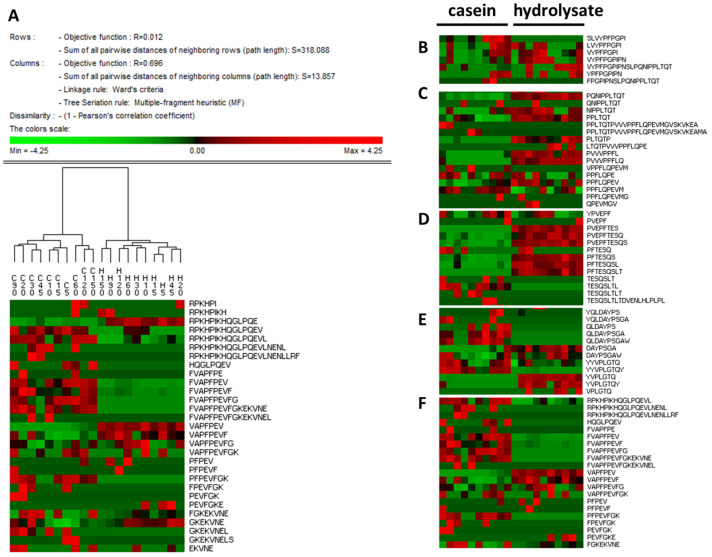
(**A**) Heat map representation (fragment) of the hierarchical clustering analysis of peptides identified in pig duodenal digests (*n* = 6) at all collected times using the Permut software. Columns correspond to collected times for substrates casein (C) and hydrolysate (H). Selection of areas from the complete representation in panels (**B**–**F**). The complete high-resolution image can be found in .png format in Appendix A.

**Figure 6 nutrients-15-01065-f006:**
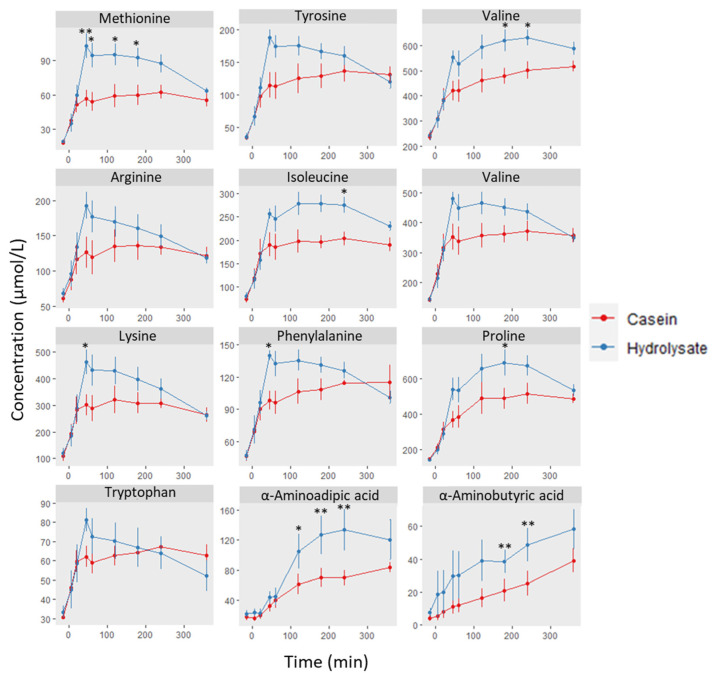
Concentration of amino acids and metabolites in plasma of pigs (*n* = 6) after intake of casein or hydrolysate over time at basal (−15 min), 5, 20, 45, 60, 120, 180, 240, and 360 min. Differences between substrates at each time point and amino acid were found through a two-way ANOVA, followed by the Bonferroni test: ** *p* < 0.01; * *p* < 0.05.

**Table 1 nutrients-15-01065-t001:** Total number of unique peptides derived from α_s1_-, α_s2_-, β-, and κ-casein identified in the duodenal effluents at each time point after intake of casein or hydrolysate, and those common among the assayed animals (*n* = 6).

Time	Casein		Hydrolysate	
	Total Peptide Number	Common Peptide Number ^1^	Total Peptide Number	Common Peptide Number ^1^
5	444	270	258	98
10	369	170	294	115
15	312	140	237	91
20	419	211	281	137
30	303	145	299	140
45	312	148	274	137
60	418	221	283	142
90	427	187	290	136
120	425	222	338	158
150	434	217	258	137
Total ^2^	3863	1931	2812	1291

^1^ In all animals (*n* = 6). ^2^ Total number of peptides and common peptide number among all time points.

## Data Availability

Not applicable.

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
