# Peer review of "Peptidomic Characterization and Amino Acid Availability after Intake of Casein vs. a Casein Hydrolysate in a Pig Model"

_nutrients, 2023, doi:10.3390/nu15051065_

Round 1
Reviewer 1 Report
This study compared the digestive behavior of casein and a casein hydrolysate in a pig model using peptidomic. The design is well-organized and the results were well-presented. Still, some problems need to be addressed.
1. Line 87: There was a missing blank space between number and h.
2. Please provide detailed information on how SDS-PAGE was performed.
3. Line 126: solvent A
4. The resolution of the figures should be improved.
5. Line 195: Please provide the full name of AA.
Author Response
- Line 87: There was a missing blank space between number and h.
This has been corrected.
- Please provide detailed information on how SDS-PAGE was performed.
All steps for the SDS-PAGE analysis have been added to the manuscript (lines 126-134).
- Line 126: solvent A
The change has been made.
- The resolution of the figures should be improved.
New figures with improved resolution have been added.
- Line 195: Please provide the full name of AA.
The suggested change has been made.
Reviewer 2 Report
Detailed analysis of peptides in the duodenal contents and changes in serum amino acid concentrations are being pursued after ingestion of diets containing casein and its hydrolysates using duodenal and jugular vein cannulated pigs. These data provide important information about protein digestion. It is also technically impressive, identifying a large number of peptides produced during the digestion process.
Major comments
1, The explanation of all tables and figures is decisively lacking. Please write these in legends and footnotes of each Tables and Figures.
2, The characters in Figs. 3-6 are blurred and unreadable. Authors need to improve significantly.
3, There is no indication of the number of pigs used. It is unclear how many pigs the results other than Fig. 1 were obtained from. These should be described in all tables and figures.
4, The authors are following the data for a short time after eating, but how did you adjust the timing of eating? Were the pigs fasted before the test?
5, Figure 1, L.316: 'slower gastric emptying' cannot be concluded only from Fig.1. Rather, Fig.1 seems to show 'slower small intestinal transit'. Opiate is known to slow down transit time in the small intestine.
6, Table 1: Correct the second line. For example, “Total peptide number found”, “Common peptide number found in all animals”. Show “Common peptide number among all time point” in the bottom line, if possible.
7, Were peptides derived from casein or its hydrolysates detected in serum? Please show the data if available.
Minor comments
1, L.21-22: The sentence following ‘suggesting’ is difficult to understand.
2, L.33-35: This part should be rewritten.
3, L,136, L.146: What is ‘R script’? Explain.
4, Figure 1: Is the label of vertical axis correct? Converting amount of nitrogen to the amount of protein (x6.25), it will exceed 1 g. Insert ‘dry’ between ‘g’ and ‘digest’ in the vertical axis label. Are error bars on the graph SD?
5, L.215-217: Show location of theses identified peptides on Fig. 3B.
6, L.250-256: This part is hard to understand. Please clearly describe what the PCI analysis in Fig. 4 indicates.
Author Response
Major comments
1, The explanation of all tables and figures is decisively lacking. Please write these in legends and footnotes of each Tables and Figures.
The legends and footnotes of all tables and figures have been revised and completed.
2, The characters in Figs. 3-6 are blurred and unreadable. Authors need to improve significantly.
Figures 3-6 have been carefully modified to improve their clarity.
3, There is no indication of the number of pigs used. It is unclear how many pigs the results other than Fig. 1 were obtained from. These should be described in all tables and figures.
The number of pigs has been added in Materials and Methods and in the table headings or figure captions.
4, The authors are following the data for a short time after eating, but how did you adjust the timing of eating? Were the pigs fasted before the test?
Test meals, casein and the casein hydrolysate (129 g of powder, on the basis of protein content), were reconstituted in water (250 mL), and offered to overnight fasted pigs for 10 min. This information was included in the Animal Experiment section.
5, Figure 1, L.316: 'slower gastric emptying' cannot be concluded only from Fig.1. Rather, Fig.1 seems to show 'slower small intestinal transit'. Opiate is known to slow down transit time in the small intestine.
The slower gastric emptying for casein is reflected in the higher protein content found in casein duodenal digests at 90 and 150 min. We agree with the reviewer that the faster gastric emptying expected for the hydrolysate at short time points is not clear and this is attributed to the sample processing (freeze-drying). This part has been clarified in the revised version of the manuscript by describing “the emptying of intact casein over a more extended period of time compared to casein hydrolysate”. On the other hand, small intestinal transit cannot be determined with the collection of effluents with a cannula at the earlier segment from the duodenum.
6, Table 1: Correct the second line. For example, “Total peptide number found”, “Common peptide number found in all animals”. Show “Common peptide number among all time point” in the bottom line, if possible.
The suggested modification of the table has been made, and a footnote has been added.
7, Were peptides derived from casein or its hydrolysates detected in serum? Please show the data if available.
Although this would provide very interesting information, unfortunately peptides in serum were not analyzed.
Minor comments
1, L.21-22: The sentence following ‘suggesting’ is difficult to understand.
The sentence has been modified.
2, L.33-35: This part should be rewritten.
The paragraph has been rewritten.
3, L,136, L.146: What is ‘R script’? Explain.
The meaning of R script has been explained in the “Materials and Methods” section.
4, Figure 1: Is the label of vertical axis correct? Converting amount of nitrogen to the amount of protein (x6.25), it will exceed 1 g. Insert ‘dry’ between ‘g’ and ‘digest’ in the vertical axis label. Are error bars on the graph SD?
We agree with the reviewer; the units have been corrected. The error bars indicate the standard error of the mean SEM.
5, L.215-217: Show location of theses identified peptides on Fig. 3B.
Figure 3B indicates the residue number on the X-axis.
6, L.250-256: This part is hard to understand. Please clearly describe what the PCI analysis in Fig. 4 indicates.
A sentence indicating what PCA explains was added. In addition, the PCA explanation in materials and methods section has been expanded.
